# Rack-and-Pinion Displacement of an Intramedullary Pin During Minimally Invasive Plate–Rod Osteosynthesis of the Canine Femur—A Case Report

**DOI:** 10.3390/ani15192777

**Published:** 2025-09-23

**Authors:** Daniel J. Wills, Max J. Lloyd, Kristy L. Hospes, William R. Walsh

**Affiliations:** 1Coast OrthoVet, Terrigal, NSW 2260, Australia; 2Surgical and Orthopaedic Research Laboratories, Clinical School of Medicine, UNSW Sydney, Sydney, NSW 2033, Australia; max.lloyd@unsw.edu.au (M.J.L.); w.walsh@unsw.edu.au (W.R.W.)

**Keywords:** MIPO, intramedullary pin, plate–rod, locking screw, rack-and-pinion

## Abstract

Intramedullary pins and plates are commonly used in combination to repair fractures in small animals. This paper discusses a technical complication not seen before, during minimally invasive repair of a femoral fracture in a dog using a common fracture fixation method. Following confirmation of pin placement, placement of the plate and screws resulted in unexpected displacement of the intramedullary pin, necessitating immediate revision. A retrieval analysis of the pin was performed, revealing surface indentations that were characterised and replicated in the laboratory. Displacement of the pin was hypothesised to have occurred due to friction between the drill bit and screws and the pin, causing a “rack-and-pinion”-like effect. The mechanism was replicated using a clinically relevant bone surrogate model. This novel failure method is of importance to veterinary orthopaedic surgeons utilising minimally invasive surgical techniques. By increasing the understanding of the techniques and possible complications that may occur during such repairs, clinical outcomes for patients can be improved.

## 1. Introduction

Conventional osteosynthesis techniques adhere to the principle of anatomic reduction with rigid fixation, providing mechanical stability to achieve primary bone healing [1]. Precise anatomic reconstruction necessitates an open surgical approach and direct reduction in fracture fragments, inherently disrupting the fracture environment and extraosseous blood supply [2,3], which may prolong time to clinical union and increase the risk of infection [4,5]. In contrast, biological osteosynthesis aims to reduce injury to the extraosseous blood supply and fracture hematoma by utilising indirect reduction methods and bridging fixation constructs to achieve relative fracture alignment and stability, producing an environment conducive to secondary bone healing [6,7].

Minimally invasive plate osteosynthesis (MIPO), characterised by indirect fracture reduction and percutaneous plate application through isolated skin incisions remote to the fracture site, adheres to the principles of biological osteosynthesis and is well described for use in dogs and cats [8]; furthermore, comparable or faster fracture healing times to those of open reduction and internal fixation have been observed in tibial and radius and ulna fractures [9,10,11].

Biologically, the use of a locking compression plate (LCP), as opposed to a dynamic compression plate, which is not compressed to the surface of the bone, aims to minimise injury to the periosteal blood supply, enhancing healing [1], making it useful for minimally invasive plate osteosynthesis (MIPO) [8]. Fixation constructs, composed of a bone plate in combination with an intramedullary pin, form a “plate–rod” construct [12]. From a technical perspective, use of an intramedullary pin (IMP) facilitates indirect reduction and assists in distraction of contracted soft tissues [13], while mechanically it serves to protect the plate against bending moments, increasing stiffness [14] and increasing the fatigue life of the construct 10-fold [15]. Pin interference during locking screw insertion, however, is a known technical consideration in plate–rod constructs due to the fixed trajectory of locking screws, which may cause drill bit breakage during hole preparation [16] or necessitate omission of the screw or placement of a monocortical screw [17].

We present a clinical case of axial displacement of an IMP during placement of a locking plate construct using a minimally invasive technique, which is hypothesised to have occurred due to friction between the drill bit and screw and the IMP, causing a “rack-and-pinion”-like effect. A retrieval analysis of the implant post-removal was performed to investigate the failure mode. Characterisation of the wear patterns produced during pin conflict with the drill bit and locking screw was performed using a simple paper impression model and a clinically relevant, fibre-filled, epoxy-laminated, open-cell polyurethane foam block model. This block model was also used to replicate the technical complication observed in this case, supporting the hypothesis.

## 2. Case Description

### 2.1. Patient

A 6-year-old, desexed, female Australian Kelpie, weighing 23.5 kg, presented two days after a motor-vehicle accident. The patient underwent treatment at our hospital (Coast OrthoVet, Terrigal, NSW, Australia) following informed consent. This case description is presented following the CARE guidelines [18]. The patient was reportedly otherwise in good general health with no known comorbidities. Initial assessment, including physical examination and thoracic- and abdominal-focused ultrasound with sonography for trauma (TFAST/AFAST) (iQ+ Vet, Butterfly, MA, USA), and radiographs, showed a closed, mid-diaphyseal transverse fracture with a caudal butterfly fragment of the left femur. The left hip had marked osteoarthritic changes. No additional injuries were detected. Pre-operative packed cell volume and total protein were within the normal reference range.

### 2.2. Anaesthetic and Analgesia

Premedication consisted of 3 µg/kg of medetomidine (Zoetis Pty Ltd., Parsippany-Troy Hills, NJ, USA) and 0.4 mg/kg of methadone (CEVA Animal Health Pty Ltd., Libourne, France) administered intramuscularly. General anaesthesia was induced by intravenous administration of propofol (10 mg/mL) (B. Braun Pty Ltd., Melsungen, Hessen, Germany), administered slowly to effect. A size 9.5 endotracheal tube was placed, the cuff was inflated, and anaesthesia was maintained with inhalational isofluorane in 100% oxygen. Peri-operative analgesia consisted of a ketamine (Randlab Pty Ltd., Revesby, Australia) constant-rate infusion (CRI) (0.5 mg/mL intravenous bolus followed by a CRI of 0.12–1.2 mg/kg/hr to effect). Regional anaesthesia was provided using femoral and sciatic nerve blocks (bupivacaine 0.5% (Pfizer, New York, NY, USA), with 1 µg/mL of medetomidine), placed using ultrasonic guidance. Cefazolin (22 mg/kg) (AFT Pharmaceuticals Pty Ltd., Takapuna, Auckland, New Zealand) was administered intravenously approximately 60 min prior to incision and repeated every 90 min throughout the procedure.

### 2.3. Surgical Technique

The skin of the left hindlimb was aseptically prepared using a standard limb-hanging technique, and adhesive drapes were applied to establish the surgical field, allowing for manipulation of the limb. Two skin incisions were made, approximately 5 cm in length, over the proximal and distal thirds of the left femur, leaving the intermediate skin intact. This approach was continued through the fascia lata along the cranial edge of the biceps femoris, which was retracted caudally to expose the femur. Bone-holding forceps were placed through these soft-tissue windows, allowing for manipulation of the fragments, which were reduced by means of forceps manipulation and normograde insertion of a 4 mm Steinmann pin introduced through the trochanteric fossa. Intra-operative radiographs were obtained using a hand-held dental X-ray unit (Port-X II, Genoray, Seoul, Republic of Korea) with dental film placed in a sterile shroud to confirm the appropriate placement of the IMP in the metaphysis of the distal fragment. The trajectory of the primary beam was perpendicular to the long axis of the bone, determined by the surgeon and assistant working in tandem. Surgical staff wore appropriate lead shielding, and additional staff left the theatre during exposure.

Following reduction, a 12-hole, 3.5 mm LCP (LCP, Synthes, Paoli, PA, USA) was applied in bridging mode. The plate was pre-contoured based on the pre-operative X-ray image of the contralateral limb, introduced through the distal window, and tunnelled through the soft tissues adjacent to the bone. Locking screws were placed in the proximal four and distal three holes of the plate, leaving the central five holes unfilled. The distal and proximal-most screws were placed first, and then the remainder, alternating proximally and distally. The IMP was cut at the level of the greater trochanter following plate placement. Surgical site closure was routine.

### 2.4. Post-Operative Imaging

Post-operative mediolateral and craniocaudal radiographic projections were acquired. These images showed adequate reduction with appropriate fragment alignment. The distal screws and two proximal screws adjacent to the fracture were bicortical. The proximal two screws were unicortical, and the second screw was inserted past the IMP. The distal tip of the IMP was located approximately 50 mm proximal to the location, documented by intra-operative radiography (Figure 1).

### 2.5. Revision

The patient was returned to theatre and prepared in a similar manner. The IMP was removed and replaced with a new pin of equal diameter. Insertion position was gauged using tactile feedback of the distal pin engaging the distal metaphysis. Repeated radiographic projections showed appropriate IMP placement (Figure 2). The final fixation construct was assessed as appropriate and satisfied the minimum recommendation for bridging fixation using a locking compression plate, i.e., ≥2 screws or ≥4 cortices in each fragment [19]. Total surgery time, including revision, was 105 min.

### 2.6. Post-Operative Management

The patient recovered with active heating until normothermic. The wound was dressed with a non-adhesive absorbent dressing and held in place with a thin, malleable, adhesive secondary dressing. Cold therapy was applied to the limb for 15 min immediately following surgery, followed by repeated applications at 6–8 hourly intervals for the first 24 h.

Post-operative medications included cephalexin 600 mg (Apex Laboratories Pty Ltd., Somersby, Australia) every 12 h for five days and carprofen 50 mg (Apex Laboratories Pty Ltd., Somersby, Australia) once every 24 h for 14 days, and a 50 µg/hr transdermal fentanyl patch (Janssen-Cilag Pty Ltd., Beerse, Belgium) was applied to the plantar surface of the operative limb and covered with an adhesive dressing.

### 2.7. Patient Follow-Up

The owner was contacted by phone 5 months post-operatively. The owner reported a return to normal activities. Follow-up radiographs were not available.

### 2.8. Implant Retrieval Analysis

The retrieved Steinmann pin was carefully examined following removal with stereo zoom microscopy (M125C, Leica Microsystems Pty Ltd., Wetzlar, Germany) to examine surface appearance compared to an unused Steinmann pin. The image was imported into Image J 1.54g [20] and calibrated relative to the scale bar embedded in the image. Length was designated as the plane along the axis of the pin, while width was designated as the perpendicular plane on the surface of the pin. Image J was used to measure features present in the pin. With repetitive features, measurements of five representatives of each feature were taken and presented as mean (±SD) in mm.

Several wear patterns were observed on the retrieved pin (Figure 3). Three vertical columns of indentations were visible on the surface of the pin, consisting of embedded indentations in a repeating pattern. At the base of these columns and on the trocar point of the pin were transverse, elliptical wear marks with vertical striations. Measurements of the wear patterns are presented in Table 1. No corrosion was evident on either the new or removed pin. Helical wear lines, varying in frequency, were also observed along the length of the pin.

The transverse wear mark at the base of columns 1 and 2 measured 1.02 × 0.92 mm and 1.01 × 0.96 mm at the base of column 3 (length × width). The total length of columns 1 and 2 was 11.86 mm, while the length of column 3 was 8.10 mm. The distal end of columns 1 and 2 was located at 70 mm proximal to the tip of the pin, and the distal end of column 3 was located at 77 mm along the pin.

### 2.9. Wear Pattern Replication—Paper Impression Model

To replicate the wear pattern on the pin, imprints of a 2.8 mm, 2-fluted standard orthopaedic drill bit (VI, Sheffield, UK) and a 3.5 mm AO locking screw (Synthes, Paoli, PA, USA) were created by placing the object against a sheet of paper and applying firm pressure while rolling along the circumference for several rotations. A lead pencil was then used to lightly shade the surface of the paper, accentuating the imprint left by the bit and screw. The drill bit and the screw were held at one of two angles, either approximating contact with the tip (approximately 45° to the paper) or the shaft (laid flat to the paper) of the bit or screw on a perpendicular trajectory relative to the longitudinal axis of the IMP (Figure 4). Image J was used to measure five instances of each feature, which are presented as the mean (±SD) in mm.

The screw produced paired horizontal indentations, and vertical striations within these pairs, corresponding to the cutting flute, relief, and threads present at the tip, are visible in Figure 4. There was an additional indentation angled at approximately 45 degrees to the horizontal indentation, corresponding to the cutting edge of the unthreaded tip of the screw. The screw shaft produced parallel striations at approximately 2 degrees to vertical, corresponding to the threads. The drill bit tip produced paired lines angled at approximately 20 degrees to the horizontal, connected by a curve, representing the leading and trailing edges of the cutting lip and land (i.e., the surface of the bit between each flute) and the margin of the drill point (i.e., the junction of the land and the relief of the drill point). The edges of the land were also visible when the shaft of the bit was imprinted. For the screw, the distance between paired indentations or between singular or pairs of indentations was approximately twice that visible on the IMP, while those made by the drill bit were almost four times greater. Spacings of vertical striations made by the screw were approximately 4–5 times greater than those visible on the pin. Measurements of the imprints are presented in Table 1, alongside those indented on the pin.

### 2.10. Laminated Polyurethane Foam Benchtop Model

Conflict between the drill bit and screw consistently resulted in proximodistal linear motion of the IMP. The direction of pin motion was determined by the direction of rotation of the screw and the side of the pin in contact with the screw. For example, a clockwise rotating screw in contact with the right side of the IMP, when viewed from the driver, resulted in proximal linear motion, while the contact of a clockwise rotating screw with the left aspect of the IMP resulted in distal linear motion. Wear patterns produced by the drill bit and screw conflict with the pin were similar to those present on the clinically retrieved pin and those produced by a simple paper impression, shown in Figure 4. A laminated block of polyurethane foam (Sawbones^®^, Pacific Research Laboratories, Vashon, WA, USA), measuring 14 × 40 × 43 mm, was used as a benchtop model to demonstrate the rack-and-pinion-like mechanism by which IMP displacement occurred in this clinical case. The block was composed of 5 PCF and open-cell rigid foam and laminated on two surfaces with fibre-filled epoxy, simulating the bone structure of the metaphyses. The laminated surfaces were placed to form the medial and lateral cortices, with the non-laminated surfaces on the top and bottom (Figure 5). With the block held in a table vice, a 4 mm Steinmann pin was inserted vertically into the block in the plane measuring 40 mm width, simulating normograde insertion of the pin in the metaphysis of a long bone. A 3.5 mm LCP was applied to the laminated surface in a standard fashion. The plate was placed such that the trajectory of the screws was directed towards the Steinmann pin, creating conflict between the drill bit/screws and the pin during construct application. The pin was examined using stereo zoom microscopy following both drill bit conflict and screw conflict. Video of the replication is provided in the Appendix A.

### 2.11. Three-Dimensional Modelling

Computer-aided design (CAD) modelling software (SolidWorks 2021 SP2.0, Dassault Systémes, France) was used to create three-dimensional renders of a generic rack-and-pinion mechanism and demonstrate this mechanism, as enacted by a self-tapping locking screw upon an IMP. Screw geometrical features, including tip, cutting flute, thread, core, and locking head, were measured using vernier callipers, and these features were modelled to scale.

## 3. Discussion

In this clinical case, friction between the rotating drill bit and screw in contact with the IMP is hypothesised to have resulted in the proximal motion of the pin. This motion can be compared to a rack-and-pinion effect, much like that seen in the steering linkage mechanisms of motor vehicles and molecules in nature [21,22]. In this case, contact between the drill bit and screw was demonstrated by the retrieval analysis, showing indentations in the pin with similar characteristics to those modelled using a simple paper impression model and by replication of these wear patterns during benchtop modelling. This interesting phenomenon was replicated in a fibre-filled epoxy laminated, open-cell polyurethane foam block model.

Two overarching strategies for the treatment of long-bone fractures have been widely discussed in the literature over at least the last 30 years, i.e., either anatomical reduction with rigid fixation or biological osteosynthesis [1,17]. While certain scenarios may be suited to an open approach, such as intra-articular fractures requiring accurate reduction or transverse, compressible, load-sharing fractures, cadaveric studies of humans and dogs show less disruption of local vasculature with MIPO approaches compared to ORIF [3,23], and favourable clinical outcomes are reported in small animals [9,10,11,12].

Implant retrieval analysis has a long history, particularly in the evaluation of failed human arthroplasty surgery [24,25], of providing surgeons and engineers deeper insight into device performance and clinical outcomes. Structural evaluation of retrieved implants has been used to identify patterns of implant failure, guiding refinement of bone plates and screws for future use [26,27,28,29]. In the veterinary literature, similar analyses have been performed with total hip replacement prostheses and with explanted cast-steel tibial plateau-levelling osteotomy plates associated with neoplasia [30,31,32]. The present analysis, however, differs in that it was performed following immediate retrieval to shed light on the mechanism behind a technical complication experienced during fracture repair using the MIPO technique.

Multiple fixation methods are available for long-bone fracture repair, including compression plates, locking plates, rods, locked intramedullary nails, and external skeletal fixators, each with its pros and cons, and in many cases, fractures may be amenable to repair by more than one option [17]. In this case, the presence of a caudal butterfly fragment indicated a need for robust bridging fixation, of which plate–rod was the most accessible and familiar option available.

The addition of a rod to a plate repair increases the stiffness of the construct, with the rod increasing in stiffness related to its radius raised to the fourth power [1]. While an IMP of 40% of the width of the intramedullary cavity is recommended in plate–rod repairs utilising locking plates based on in vitro biomechanical testing to maximise construct stiffness [15], this does not necessarily consider friction between the bone and components of the fixation construct. It is not uncommon to encounter some contact between the IMP and locking screws during plate placement, due to the fixed trajectory of the screw [19,33]. Contact between implants and/or instruments may be accompanied by tactile and/or auditory feedback [34]. Indeed, sufficient contact may result in drill bit breakage due to three-point bending [16] and is reported in the human literature, although the true incidence may be underreported [16,35]. It may be that friction between an IMP and the metaphyseal bone contributes to the stability of the repair construct and that, once placed, friction between contacting screws and intramedullary rod creates additional friction in the system, serving to increase the overall stability of the construct. Despite being of appropriate diameter in this clinical case, there was insufficient friction between the IMP and the bone in which it was seated, resulting in displacement during screw placement. Further, upon revision, a new pin of equal diameter was replaced at the required depth in the distal metaphysis, unhindered, without removal of the bone plate or screws. This could reflect a low volume and/or density of trabecular bone in the metaphyses in this patient, providing minimal resistance to deflection, along with the intentionally small diameter (40% of the intramedullary canal) of the pin selected, allowing placement to progress beyond the screws left in place using power insertion.

Post-operatively, the mediolateral radiograph showed the IMP was located cranial to the screws. During screw insertion, by virtue of the thread geometry, clockwise rotation of the screw resulted in proximal axial displacement of the IMP. Figure 6 shows a three-dimensional rendering of the screw and the IMP enacting the rack-and-pinion-like mechanism, demonstrating the failure mode. The location of the wear patterns along the length of the pin corresponds with the region of the pin adjacent to the three proximal screws.

Wear debris was not visible in the post-operative radiograph; however, it may have been deposited within the bone or surrounding tissue during contact between the drill or screw and IMP. Multiple animal studies have shown a negative tissue response to steel wear debris, resulting in inflammation and osteolysis, which could complicate healing or result in implant loosening [36,37,38].

While the pin was not examined for indentations prior to placement, it was previously unused. Comparison of the indentations in the pin with those in both the impression and benchtop replication models suggests that they were caused by conflict with the tip and shaft of the drill bit and screw during plate application. Variables including contact angle, forces, rotation speed, curvature of the IMP, and implant material properties (e.g., modulus, relative hardness of the pin and screw) could potentially influence the contact wear pattern by varying the cutting forces [39,40]. Differences in spacing were present between modelled indentations and those seen on the retrieved pin, which could be explained by differing forces and angles present between the two scenarios.

A power driver was used to remove the IMP and insert the revision pin. Similar to screw threads, in which varying thread pitch influences the rate of axial advancement in the bone, rotation of the pin with varying axial traction during extraction is consistent with the variably spaced helical wear marks along the length of the pin. Power drivers improve the ergonomics and efficiency of orthopaedic surgery but may impart large amounts of energy to the bone. This energy is primarily converted into heat, which may cause thermal necrosis with sufficient exposure peaks and durations [41,42]. The threshold for thermal necrosis is commonly cited as 47 °C for 1 min based on landmark in vivo studies performed by Eriksson and Albrektsson (1983, 1984) and should be considered by the surgeon utilising power in orthopaedic surgery [43,44].

MIPO may be facilitated by intra-operative radiographic [45] or fluoroscopic imaging [46,47]. Personnel and patient exposure to radiation, however, is a concern during minimally invasive surgery and should be minimised where possible [47,48]. Indeed, MIPO has been advocated as a repair option for appropriate long-bone fractures in dogs and cats without using intra-operative imaging; however, failure to capture the distal fragment with the IMP due to extraosseous placement was a relatively frequent complication with 3/14 (21%) plate–rod cases, requiring immediate revision [49]. In this case, a hand-held dental X-ray unit was used to facilitate intra-operative imaging. Although image resolution may be limited due to a low radiation dose and the relatively large volume of the limb being imaged, this mode offers a low-radiation [50,51,52] intra-operative radiographic imaging option in our orthopaedic practice. Indeed, dental radiography is commonly available in many veterinary hospitals. Its availability and highly mobile nature, in combination with low radiation output, may allow it to serve as a useful imaging method when other imaging modalities, such as fluoroscopy, are not available.

A limitation of intra-operative imaging, whether with hand-held radiography or fluoroscopy, is the limited field of view, which may result in the omission of critical information unless additional views are taken, subjecting the patient and staff to additional radiation doses [53]. In the present case, only a mediolateral image was taken to confirm the distal seating of the IMP. While two-plane orthogonal views would be required for the surgeon to discern whether the rod was captured within the intramedullary canal [49], the purpose of this image was to identify the tip of the pin at the appropriate proximodistal level within the distal metaphysis, rather than intramedullary capture, which was apparent during reduction in the primary bone fragments. Hence, one view was sufficient based on a perpendicular trajectory determined by the surgeon and assistant working in tandem. While additional operator and patient radiation exposure and a lengthening procedure time with repeat intra-operative imaging should be avoided where possible [54], additional imaging in this case may have eliminated the need for immediate revision and increased anaesthetic and surgical time associated with revision and additional post-revision imaging. During open procedures, pin insertion depth may be directly measured against the bone using a ruler or pin of similar length [55]. In the MIPO scenario, however, only limited regions of the bone are directly viewed, limiting the accuracy of such techniques. In this case, the IMP was cut at the depth of the greater trochanter after plate placement, such that displacement of the pin was not obvious, as it would have been if it were cut prior to plate placement. This case demonstrates that balancing radiation dose, imaging field of view, and surgical times must be considered by the surgeon.

In addition to the technical aspects of this case, the fixation of the IMP in the proximal and distal bone warrants further investigation. A fibre-filled, epoxy-laminated, open-cell rigid foam block was used as a model to demonstrate the mechanism by which axial displacement of the intramedullary rod occurred. The open-cell portion of the block has a density of 5 PCF, representing low-density or osteoporotic bone, while the epoxy laminate contains short glass fibres to simulate cortical bone for structural testing of fixation devices [56]. The density of the metaphyseal bone in the healthy canine femur varies but has been reported in the range of 60–100 PCF [57,58]. Polyurethane foam block models do not replicate the anatomy of the canine femur, nor does the structure of the foam accurately represent trabecular bone structure [59]. Ex vivo models lack marrow, blood supply, and soft tissues and may not accurately represent the forces present clinically [60]. Hence, the benchtop replication demonstrated the mechanism by which an IMP may undergo axial displacement within metaphyseal bone, but it may not replicate the frictional conditions present clinically. In a plate–rod construct, the plate resists torsion, bending, and compressive forces, while the rod adds to the bending stiffness [61]. The displacement of the rod in this case, however, suggests that the rod may not contribute clinically to axial compression. This may be of relevance when planning or evaluating fracture fixation options or post-operative healing patterns.

As a case report, this study is limited in that the bone density of the femur and insertion forces during implant placement were not quantified. In addition, the frequency of pin displacement due to the rack-and-pinion mechanism is unknown. Further in vitro or ex vivo studies would be of use to more precisely characterise the conditions under which this phenomenon may occur. Further clinical observation is warranted to determine the rate at which this mechanism occurs during application of similar fixation constructs. Finally, while follow-up at 5 months indicated acceptable clinical outcome by owner assessment, no other outcomes such as follow-up imaging were available.

## 4. Conclusions

In the present case, appropriate IMP placement was documented using intra-operative imaging; however, subsequent events resulted in an axial shift in IMP position, necessitating immediate revision. The potential for significant axial displacement of the IMP in plate–rod fixation constructs due to conflict with drill bits and screws during construct application is possibly caused by a rack-and-pinion-like effect, and surgeons should confirm implant positioning if implant conflict is recognised intra-operatively.

## Figures and Tables

**Figure 1 animals-15-02777-f001:**
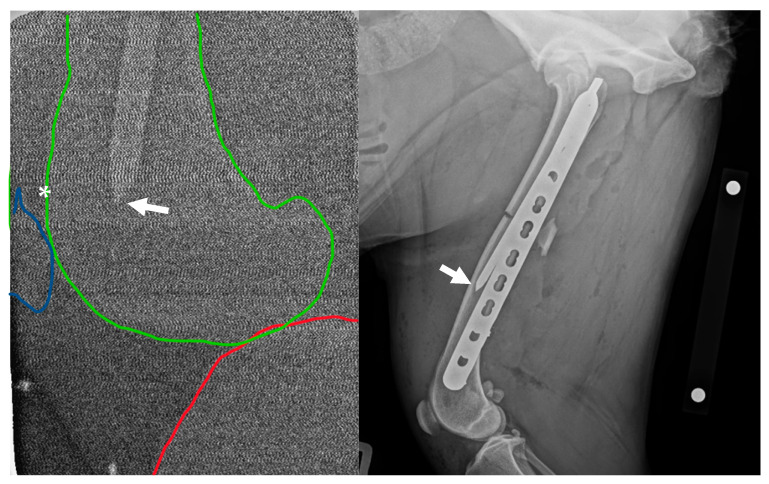
**Left**: Intra-operative low-radiation mediolateral radiograph acquired using a hand-held dental X-ray unit. The distal tip of the intramedullary pin (arrow) is apparent at the level of the femoral metaphysis/base of patella (asterisk). The distal femur is outlined in green, the proximal tibia is outlined in red, and the patella is outlined in blue. **Right**: Initial post-operative mediolateral view showing insufficient purchase of the distal fragment with the intramedullary pin; the tip of the pin (arrow) is located in the distal-third of the diaphysis, approximately 50 mm proximal to the patella. NB a calibration marker is present, with a distance of 10 cm between the centre of the metal spheres.

**Figure 2 animals-15-02777-f002:**
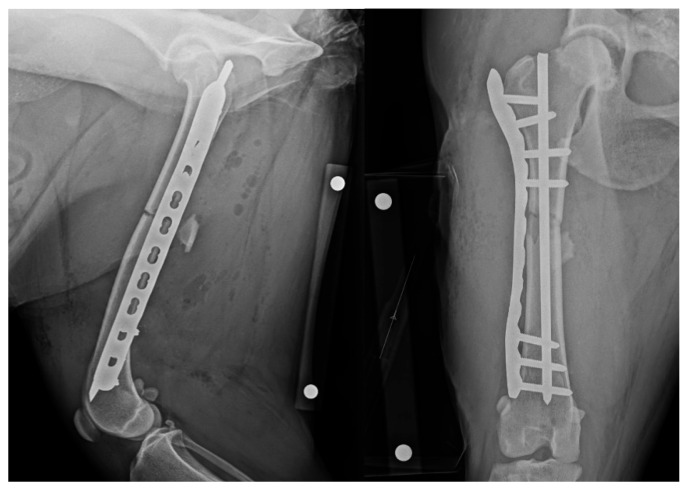
Revision post-operative mediolateral and craniocaudal radiographic projections with revised appropriate intramedullary pin length. NB a calibration marker is present, with a distance of 10 cm between the centre of the metal spheres.

**Figure 3 animals-15-02777-f003:**
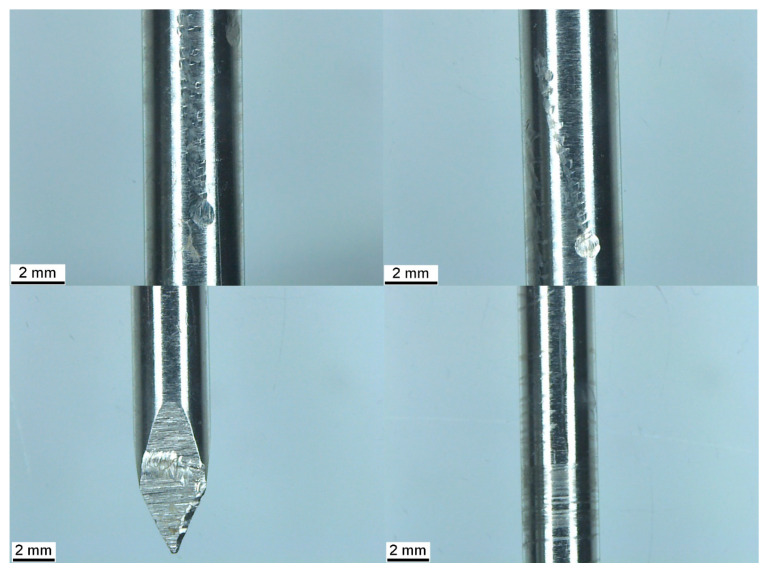
**Top left and right**: Three columns of horizontal indentations with a larger elliptical wear mark at the base of each. **Bottom left**: Transverse wear mark with vertical striations. **Bottom right**: Helical wear marks along the length of the pin.

**Figure 4 animals-15-02777-f004:**
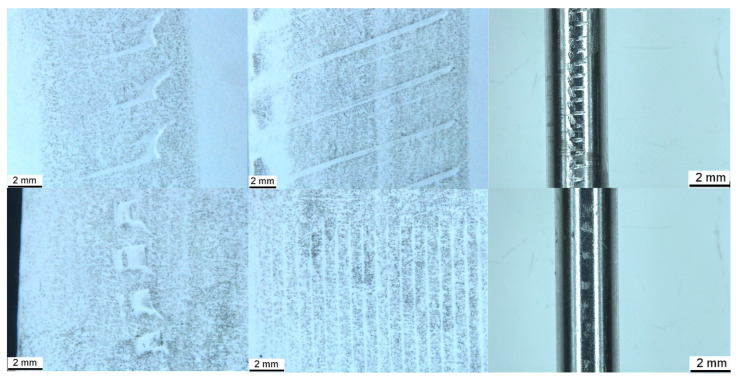
Imprints of 2.8 mm standard orthopaedic drill bit tip (**top left**), shaft (**top centre**), and 3.5 mm AO locking screw tip (**bottom left**) and shaft (**bottom centre**). Similar imprints are visible on the pin retrieved from the benchtop model (top and bottom right).

**Figure 5 animals-15-02777-f005:**
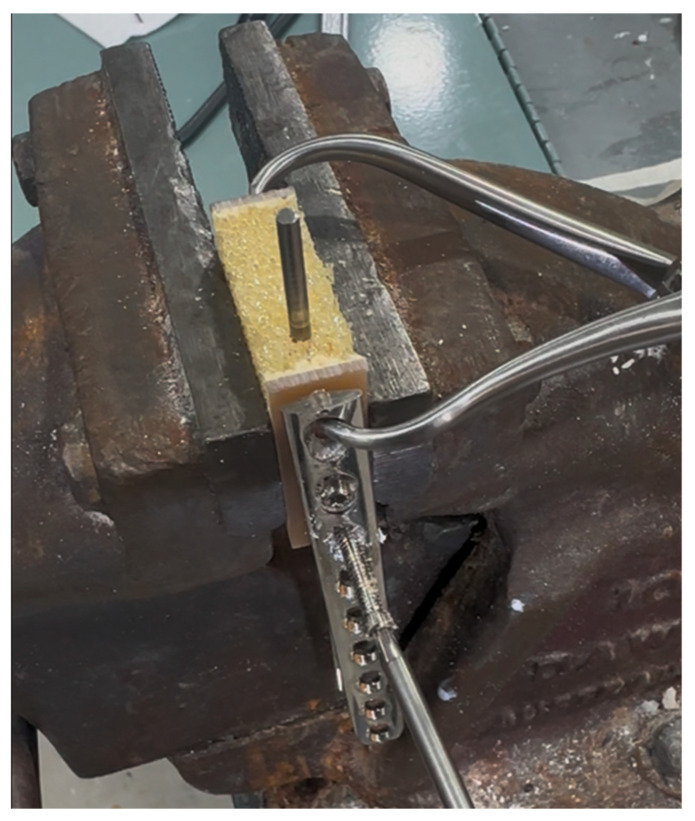
Benchtop replication of the failure mode. A block of fibre-filled, epoxy-laminated, 5 PCF open-cell rigid polyurethane foam (Sawbones^®^ Pacific Research Laboratories, Vachon, WA, USA) was held in a table vice. The laminated surfaces formed the lateral and medial cortex. A 4 mm Steinmann pin was inserted in a proximodistal orientation, within the open-cell foam, replicating the clinical scenario of an intramedullary rod in metaphyseal bone. A 3.5 mm LCP was applied, with screws in a mediolateral trajectory. Contact between the rotating drill bit and the screw during plate application resulted in linear motion of the pin.

**Figure 6 animals-15-02777-f006:**
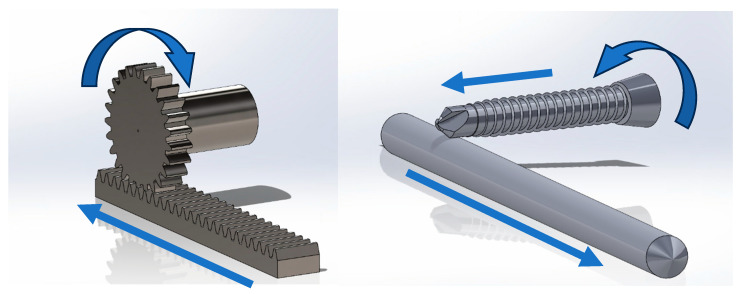
**Left**: Rack-and-pinion mechanism. Rotation of the pinion results in corresponding linear motion of the rack. **Right**: A screw (or drill bit) contacting an intramedullary rod may act as a pinion if friction between the screw and pin is greater than friction between the pin and the bone in which it is seated. The direction of linear motion of the intramedullary pin is dependent on the direction of rotation of the screw and the side of the pin in contact with the screw. Arrows denote direction of rotation and linear motion of bodies in the system.

**Table 1 animals-15-02777-t001:** Measurements of 2.8 mm drill bit and 3.5 mm locking screw impressions visible on retrieved pin and on paper impressions made with the tip (45° angle) or shaft of the drill bit and screw. Corresponding geometrical features are noted in parentheses.

	Column 1 and 2—Retrieved Pin	Column 3—Retrieved Pin	Screw Tip Impression	Screw Shaft Impression	Drill Bit Tip Impression
Distance between paired indentations (i.e., cutting flute relief width)	0.466 (±0.114)	Singular	1.19 mm (±0.166)	Not present	1.74 mm (±0.053)
Distance between singular indentations or pairs (i.e., cutting flute width)	0.611 (±0.115)	0.603 (±0.086)	1.369 mm (±0.118)	Not present	2.29 mm (±0.229)
Distance between vertical striations	0.070 (±0.042)	0.096 (±0.018)	0.400 mm (±0.0965)	0.548 mm (±0.033)	Not present

## Data Availability

Non-identified data presented in this study are available upon request from the corresponding author.

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
