# Peer review of "Rack-and-Pinion Displacement of an Intramedullary Pin During Minimally Invasive Plate–Rod Osteosynthesis of the Canine Femur—A Case Report"

_animals, 2025, doi:10.3390/ani15192777_

Round 1

Reviewer 1 Report (Previous Reviewer 1)

Comments and Suggestions for Authors

Dear Authors,
Thank you for your valuable corrections.
Please ensure that the limitations regarding the need for broader clinical observations of the rack-and-pinion phenomenon or the determination of its frequency in post-mortem studies are clearly emphasised in the Discussion section.

You should also mention the potential for the drill to break upon contact with a nail in the introduction, before discussing this in more detail in the 'Discussion' section. Although the veterinary literature is limited, the human literature on intraoperative drill bit breakage is extensive.

Author Response

Dear Authors,
Thank you for your valuable corrections.
Please ensure that the limitations regarding the need for broader clinical observations of the rack-and-pinion phenomenon or the determination of its frequency in post-mortem studies are clearly emphasised in the Discussion section.

To more completely address the reviewer’s comment, we have subtracted from line 431-432: “and further work is required prior to use of this model for direct clinical translation of results”

And added an additional paragraph has been added to line 423 - 430

“As a case report, this study is limited in that the bone density of the femur and insertion forces during implant placement were not quantified. In addition, the frequency of pin displacement due to the rack-and-pinion mechanism is unknown. Further in vitro or ex vivo studies would be of use to more precisely characterise the conditions under which this phenomenon may occur. Further clinical observation is warranted to determine the rate at which this mechanism occurs during application of similar fixation constructs.”

You should also mention the potential for the drill to break upon contact with a nail in the introduction, before discussing this in more detail in the 'Discussion' section. Although the veterinary literature is limited, the human literature on intraoperative drill bit breakage is extensive.

Thank you. Drill breakage is included as a possible consequence on line 68 of the introduction.

Added to discussion Line 317-319 :” Indeed, sufficient contact may result in drill-bit breakage due to 3-point bending (16) and is reported in the human literature, although the true incidence may be under reported (16, 32).”

Added references.

Reviewer 2 Report (Previous Reviewer 2)

Comments and Suggestions for Authors

the revision presented show an overall apreciable improvement and the suggestions of the reviewers have been effectively accepted.

only two times the term "rod" (line 92 and end of Figure2) is still maintained.

Author Response

the revision presented show an overall apreciable improvement and the suggestions of the reviewers have been effectively accepted.

only two times the term "rod" (line 92 and end of Figure2) is still maintained.

Rod has been removed and IMP used.

Reviewer 3 Report (Previous Reviewer 3)

Comments and Suggestions for Authors

Dear authors,

It has been a pleasure reviewing the revised version of your manuscript. In this version, it has improved considerably. However, there are still some important aspects that need to be addressed to bring the manuscript up to case report standards.

  • It is not common to have a section titled “3. Results” in a case report because it is not an experimental, clinical, retrospective, multiple-case series study or one with comparative groups. The most widely accepted structure (also recommended by CARE guidelines and most journals) is: introduction, case description, discussion and conclusions. In a case report, the “results” (clinical evolution, radiological findings, complications, etc.) should be integrated into the case description.
  • Although the journal Animals does not expressly mention the mandatory adherence to the CARE guidelines for case reports, they constitute a widely accepted international standard to ensure that clinical case reports are complete, transparent, and useful to the scientific comunity. I recommend that authors adapt their manuscript to the guidelines or, at least, complete and include the checklist as supplementary material, which would increase the quality and value of the report.
  • Although the inability to measure bone density or insertion strength is justified, it would be helpful to explicitly include this in the discussion section as a limitation of the case report. In general, a clear "limitations" section should be added at the end of the discussion section, listing all the limitations of the study.
  • In relation to the above, there is only information via phone call at 5 months, with no X-rays. They should recognize in the "limitations section" that the lack of follow-up images limits the evidence of clinical outcome.
  • Figure 1 is still of very limited radiological quality. A more obvious arrow or marker could be added to indicate pin displacement.
  • The abbreviations are now correctly listed, but it would be advisable to standardize their first appearance in the text (some are defined late).

Best regards.

Author Response

Dear authors,

It has been a pleasure reviewing the revised version of your manuscript. In this version, it has improved considerably. However, there are still some important aspects that need to be addressed to bring the manuscript up to case report standards.

  • It is not common to have a section titled “3. Results” in a case report because it is not an experimental, clinical, retrospective, multiple-case series study or one with comparative groups. The most widely accepted structure (also recommended by CARE guidelines and most journals) is: introduction, case description, discussion and conclusions. In a case report, the “results” (clinical evolution, radiological findings, complications, etc.) should be integrated into the case description.

Results removed, headings previously under results have been incorporated into case description and renumbered. Figures renumbered so they are listed in order of use.

  • Although the journal Animals does not expressly mention the mandatory adherence to the CARE guidelines for case reports, they constitute a widely accepted international standard to ensure that clinical case reports are complete, transparent, and useful to the scientific comunity. I recommend that authors adapt their manuscript to the guidelines or, at least, complete and include the checklist as supplementary material, which would increase the quality and value of the report.

Thank you for the suggestion. Due to the focus of this report being the surgical technique and implant retrieval analysis, we have not adopted the CARE format for the body of the manuscript, although, the manuscript does satisfy the majority of the guideline checklist. We have included the checklist as supplementary material to increase the value of the report. We have made the changes to the article format as above.

  • Although the inability to measure bone density or insertion strength is justified, it would be helpful to explicitly include this in the discussion section as a limitation of the case report. In general, a clear "limitations" section should be added at the end of the discussion section, listing all the limitations of the study.

Added to discussion a final limitations section including these limitations. Line 423-430:

“As a case report, this study is limited in that the bone density of the femur and insertion forces during implant placement were not quantified. In addition, the frequency of pin displacement due to the rack-and-pinion mechanism is unknown. Further in vitro or ex vivo studies would be of use to more precisely characterise the conditions under which this phenomenon may occur. Further clinical observation is warranted to determine the rate at which this mechanism occurs during application of similar fixation constructs. Finally, whilst follow-up at 5 months indicated acceptable clinical outcome by owner assessment, no other outcomes such as follow-up imaging were available.”

  • In relation to the above, there is only information via phone call at 5 months, with no X-rays. They should recognize in the "limitations section" that the lack of follow-up images limits the evidence of clinical outcome.

Added to discussion line 428-430:

“Finally, whilst follow-up at 5 months indicated acceptable clinical outcome by owner assessment, no other outcomes such as follow-up imaging were available.”

  • Figure 1 is still of very limited radiological quality. A more obvious arrow or marker could be added to indicate pin displacement.

The limitation of low resolution of the imaging modality is discussed (line 374-377). Figure 1 has been altered to show the immediate post-operative film next to the intraoperative film. An asterix has been added to the intraoperative film to designate the patella. An arrow has been added to both films to show the arrow. The distance moved is apparent with these images side-by-side.

  • The abbreviations are now correctly listed, but it would be advisable to standardize their first appearance in the text (some are defined late).

We could see one instance of the use of locking compression plate (LCP) after it had already been defined. This was removed, keeping the acronym (line 122)

Round 2

Reviewer 1 Report (Previous Reviewer 1)

Comments and Suggestions for Authors

Dear authors,
Thank you very much for the changes you have made to your manuscript.

Author Response

Thank you for the reviews that have improved the work.

Reviewer 3 Report (Previous Reviewer 3)

Comments and Suggestions for Authors

Dear authors,

The manuscript has improved substantially. Just a few minor notes.

In the text (usually at the beginning of the "case description" section) it should be mentioned that the CARE guidelines have been followed. 

In addition, the "supplementary material" section should be paraphrased.

Best regards. 

Author Response

Thank you, the Care guidelines sentence has been added and referenced.

The supplementary material line requires a URL, perhaps this is an editorial change - the videos have been uploaded.

Thank you for the review

This manuscript is a resubmission of an earlier submission. The following is a list of the peer review reports and author responses from that submission.

Round 1

Reviewer 1 Report

Comments and Suggestions for Authors

Dear authors,
Congratulations on making this valuable clinical observation. Although it is possible for an intramedullary pin to slip during drilling or screw insertion, it is unclear how frequently this occurs. To strengthen the scientific value of your report, I recommend performing the MIPO procedure that you described on at least 30 canine femurs (i.e. in 15 canine cadavers) and then statistically analysing the incidence of the phenomenon you describe. The experiment you performed on the polyurethane block does not reflect intraoperative conditions.

You should also update the discussion and references to include other potential complications that could arise from a drill or screw colliding with an intramedullary pin.

Please also add keywords.

Reviewer 2 Report

Comments and Suggestions for Authors

The core of the comment expressed for the revision doesn’t have to consider remarks on technical details or the opportunity to apply this configuration for the resolution of the fracture.

However, this quite simple fracture configuration could be treated in a simpler manner without complication, not forcing the solution of a  Minimally Invasive (aimed to insert a LCP) Plate-Rod Osteosynthesis mainly devoted to more complex diaphyseal fractures.

The p.o. image of the L-L or M-L projection shows an imperfect alignment of the plate with a distal screw that seems not to impinge the caudal cortex.

In the paper are not reported the possible sensation of conflicting metal/metal during the drilling that can induce to hypothesize not to have a good implant creation and the possible assessment  of the proximal climb back of the pin in the same time of drilling

31- replicated in a benchtop laminated polyurethane foam block model. And 186 A laminated block of polyurethane foam (Sawbones®, Pacific Research Laboratories, Vashon, WA) was used as a benchtop model.  Why it was choosen to apply this kind of experimental design and material and the dimension of the block were the same of the diameter of the bone to replicate this situation? Was the pin inside this block fixed up and down like it was inside the femour?

34  - “rack-and-pinion” mechanism, 75 - causing a “rack-and-pinion” effect and Figure 6. This similarity is not perfectly fitting. The mechanism indicated implies two congruent surfaces that move one to the other. In this case at least a “sort of “rack-and-pinion” mechanism” or “rack-and-pinion” like mechanism must be indicated. In fact in all the paper the possible deleterious effects of the filling coming from abrasion of the rod are not discussed.

73 – surprising clinical case. The term surprising is not appropriate for a surgeon, that must be aware of his/her work. The drill / pin conflict is possible with the configuration plate and proximal rod displacement could be an infrequent consequence of it, to be indicated in literature.  

142 The IMP was removed and replaced with a new pin and 314 Further, on revision, a new pin of appropriate diameter was replaced to the required depth in the  distal metaphysis, unhindered, without removal of the bone plate and screws. Probably better to indicate if the diameter of the new pin was the same.

Figure 2. it seems the most proximal of the distal screws isis inefficient

300 - In this case, the presence of a caudal butterfly fragment indicated a need for robust bridging fixation, of which plate-rod was the most accessible and familiar option available. Just an opinion of the rewiev: the implant option used is indicated for most of the fracture configuration but probably in this case the risk of conflict overcome the necessity for this double implant insertion.

Reviewer 3 Report

Comments and Suggestions for Authors

Dear Authors,

It was a pleasure to review your valuable manuscript. The manuscript describes an unusual technical complication never before reported: axial displacement of an intramedullary needle (IMP) due to a “rack-and-pinion” effect. I find it very interesting because of the clinical relevance it may have for fellow veterinarians dedicated to orthopedic surgery, specifically those who use the MIPO technique.

Therefore, I would like to offer some suggestions to substantially improve the quality of the article and be considered for publication.

Formatting suggestions.

  • The title should be modified to indicate that it is a case report. For example: "Case Report: Rack-and-pinion displacement of an intramedullary pin during minimally invasive plate-rod osteosynthesis of the canine femur."
  • After the summary, the keywords have not been added. Please add them.
  • I believe the format should be changed to the one typically used for this type of article (case report). Therefore, the sections should be: (1) Introduction, (2) Case description, (3) Discussion, (4) Conclusions.
  • On the other hand, I suggest that authors follow the CARE guidelines (https://www.care-statement.org/checklist). In addition, the checklist must be completed and attached as supplementary material to the manuscript. The CARE guidelines serve as a standardized guide for writing and publishing clinical case reports in a manner that is clear, comprehensive, and useful to the scientific and medical community.

Suggestions regarding content.

  • To strengthen the hypothesis of low trabecular resistance as a predisposing factor, I suggest, if possible, incorporating an assessment of bone density (e.g., by semiquantitative radiographic evaluation).
  • Was the force applied during screw insertion quantified? This could help better understand the mechanical parameters involved.
  • The quality of Figure 1 is very low, and its significance is barely perceptible. Could it be improved?
  • I also suggest improving the scale bars (two images on the right) in Figure 5. They are barely perceived.
  • Complication figures are mentioned in the discussion, but they are not always supported by direct references. Please support each critical claim with at least one primary source or reliable review.
  • Please, complete the abbreviations section. Only one is specified (IMP), but actually, there are many in the text (CAD, MIPO, LCP...).

References section.

  • Please review the bibliography carefully according to Animals guidelines.
  • For example, references 1 and 12 are the same (Lines 433 and 458).
  • Besides, the DOI is missed in some references
